# Electrophysiological and Behavioral Evidence for Hyper- and Hyposensitivity in Rare Genetic Syndromes Associated with Autism

**DOI:** 10.3390/genes13040671

**Published:** 2022-04-11

**Authors:** Anastasia Neklyudova, Kirill Smirnov, Anna Rebreikina, Olga Martynova, Olga Sysoeva

**Affiliations:** 1Institute of Higher Nervous Activity and Neurophysiology, Russian Academy of Science, 117485 Moscow, Russia; anastacia.neklyudova@gmail.com (A.N.); kirillsmirnov@ihna.ru (K.S.); anna.rebreikina@gmail.com (A.R.); omartynova@ihna.ru (O.M.); 2Sirius Center for Cognitive Research, Sirius University of Science and Technology, 354340 Sochi, Russia

**Keywords:** Fragile X syndrome, Angelman syndrome, Phelan–McDermid syndrome, Rett syndrome, neurofibromatosis type 1, tuberous sclerosis, electroencephalogram (EEG), event-related potentials (ERP), biomarkers, sensory deficits, autism spectrum disorder (ASD)

## Abstract

Our study reviewed abnormalities in spontaneous, as well as event-related, brain activity in syndromes with a known genetic underpinning that are associated with autistic symptomatology. Based on behavioral and neurophysiological evidence, we tentatively subdivided the syndromes on primarily hyper-sensitive (Fragile X, Angelman) and hypo-sensitive (Phelan–McDermid, Rett, Tuberous Sclerosis, Neurofibromatosis 1), pointing to the way of segregation of heterogeneous idiopathic ASD, that includes both hyper-sensitive and hypo-sensitive individuals. This segmentation links abnormalities in different genes, such as *FMR1, UBE3A, GABRB3, GABRA5, GABRG3, SHANK3, MECP2, TSC1, TSC2*, and *NF1*, that are causative to the above-mentioned syndromes and associated with synaptic transmission and cell growth, as well as with translational and transcriptional regulation and with sensory sensitivity. Excitation/inhibition imbalance related to GABAergic signaling, and the interplay of tonic and phasic inhibition in different brain regions might underlie this relationship. However, more research is needed. As most genetic syndromes are very rare, future investigations in this field will benefit from multi-site collaboration with a common protocol for electrophysiological and event-related potential (EEG/ERP) research that should include an investigation into all modalities and stages of sensory processing, as well as potential biomarkers of GABAergic signaling (such as 40-Hz ASSR).

## 1. Introduction

Autism spectrum disorder (ASD) is a neurodevelopmental disorder characterized by deficits in social and communicative skills, behavioral stereotypes, and sensory abnormalities. Its prevalence is one in 54 children, with the number of identified cases rising annually [1].

Atypical sensory processing is one of the key characteristics of autistic people (for review: [2,3]). Behaviorally, sensory abnormalities can present as hypo- and hyperresponsiveness (or hypo- and hyperreactivity). The Diagnostic and Statistical Manual of Mental Disorders, 5th Edition (DSM-5) included both hyper- and hyporeactivity to sensory stimulation as well as unusual interest in some sensory aspects of the environment to the diagnostic criteria for ASD [4]. These abnormalities can have a cascading influence on higher-level functions, such as language or social inference.

Despite strong evidence for a sensory deficit in ASD, its particular mechanisms are still unknown. Studies show both hypo- and hypersensitivity in children with ASD [2,5,6] and one of potential reasons for this multidirectional deficit is the high heterogeneity of idiopathic ASD (the form of autism where the biological cause of the disease is not identified). A possible way to overcome this problem is to subdivide this diverse population into subgroups with a similar sensory phenotype. Ideally, this subdivision should be based on the underlying biological mechanisms and one way to do that is to study syndromic forms of ASD—genetic syndromes with a high prevalence of autistic symptoms [7]. Currently, more than 80 of such syndromes have been identified [8]. However, electrophysiological and behavioral studies about sensory sensitivity were not available for many disorders as those are very rare conditions and are difficult to investigate. Here, we have focused on six syndromes, namely Fragile X syndrome (FXS), Angelman syndrome (AS), Phelan–McDermid Syndrome (PMS), Rett syndrome (RS), Tuberous sclerosis (TSC), and Neurofibromatosis type 1 (*NF1*), which are caused by abnormalities in *FMR1, UBE3A, GABRB3, GABRA5, GABRG3, SHANK3, MECP2, TSC1, TSC2*, and *NF1*. These genes are not only causative for these rare syndromes, but also connected with susceptibility to autistic symptoms, even without diagnostics of the syndromes themselves [9,10,11]. For example, the *SHANK3* gene is the most common gene associated with ASD and *SHANK3* mutations are reported in 0.7% of patients with ASD (2.1% if ASD is accompanied by moderate to profound intellectual disability) [12]. Moreover, the autistic brain is characterized by altered methylation patterns in *SHANK3* in ≈15% of patients as evidenced from post-mortem studies [13]. Another example is the *MECP2* gene, in which abnormalities have only been found in some rare cases of idiopathic ASD [14,15,16], but, at the same time, decreased *MECP2* expression in the cortex was characterized in about 80% of autism cases [17]. Thus, existing evidence points to the widespread implication of molecular–genetic pathways that are associated with rare genetic abnormalities being linked to ASD development. In the light of these facts, an investigation of physiological mechanisms of sensory deficits on syndromic forms can be applicable at least to some forms of ASD as well.

An addition, these syndromes have well-established animal models that allow us to delve into the biological mechanisms underlying the behavioral symptoms on lower biological levels using a large variety of experimental approaches. These models are also widely used to develop novel treatment approaches both for syndromic and idiopathic ASD [18,19].

The first step for the identification of hypo- and hyperresponsive phenotypes in ASD samples is to understand what objective measures (or biomarkers) can be used. Electroencephalography (EEG) is a promising technique for discovering biomarkers of sensory abnormalities in the ASD patient due to its good time resolution, wide-spread clinical availability, and translatability between human and animal studies. Moreover, EEG allows for the investigation of sensory processing even in non-cooperative, challenging patients who cannot be involved in active tasks due to their condition. For example, assessing the problems underlying speech delay is hardly possible using behavioral methods in non-verbal individuals; however, impairments in auditory processing evaluated using EEG, e.g., event-related potential (ERP)—EEG signals averaged over multiple segments in response to external stimuli—can be good predictors of this delay and can provide information about the functional meaning of these impairments [20].

Another concept that frames our review is the theory of excitation/inhibition imbalance (E/I imbalance) in neural circuits as a potential cause of autistic symptoms [21]. Excitation is mediated by pyramidal neurons that release glutamate as neurotransmitters, and inhibition is mediated by interneurons that release GABA. As pyramidal neurons and interneurons are interconnected, glutamatergic excitation and GABAergic inhibition affect each other and both contribute to the E/I balance [22,23]. Here, we will not dive deep into the molecular mechanisms of E/I imbalance connected with autistic symptoms in syndromes; however, biomarker detection can specify directions for investigation of different molecular paths in future research.

Overall, in this article, we will review recent findings on sensory abnormalities in different syndromes associated with autism, focusing on ERPs as a non-invasive indicator of external information processing. We will also review major findings in resting-state EEG since background EEG characteristics can underlie an individual neurophysiological response to the environmental stimuli. As syndromic forms of ASD are very rare, it is not always possible to draw conclusions only from patients’ data. Where possible, we will try to relate patients’ data with those obtained in animal studies to get a more integrated picture on the mechanisms of the disorder. We assume that this approach can help us to shape the future directions for electrophysiological studies in patients with ASD. Based on data from the literature, we divided syndromes into mostly hyper- or hyposensitive phenotypes. We structured each syndrome presentation by first providing the general information about the syndrome and its genetic etiology, followed by sensory phenotype, and then review available EEG/ERP studies, pointing to the clinical epilepsy phenotype with the major focus on quantitative analysis, finally ending with a short summary. The phenotypic descriptions of the syndromes are presented in Table 1 and the summary of electrophysiological data is presented in Table 2. The brief description of the ERP papers retrieved by the pubmed search that are included in the review can be found in the Appendix A. The paper is concluded by the general discussion of the most promising EEG/ERP biomarkers that might link behavioral and genetic research in ASD.

## 2. Hypersensitive Syndromes

### 2.1. Fragile X Syndrome

#### 2.1.1. General Info

Fragile X syndrome (FXS) is a neurodevelopmental disorder associated with intellectual disability, social deficits, hypotonia, and anxiety. Its prevalence varies from approximately 1:4000 in males to 1:8000 in females [35], making it the second most common chromosome abnormality that leads to intellectual disability after Down’s syndrome [71]. Intellectual disability is mild or moderate in this syndrome [25]. FXS has a strong comorbidity with ASD, with a prevalence of 22% for both sexes [26,35]. Another frequently associated condition is attention deficit and hyperactivity disorder (ADHD), which occurs in 60–84% of cases [31,32]. FXS patients display weaknesses across all language and literacy domains compared to peers, with particular noticeable impairment being the higher occurrence of word repetition in these patients [30]

Fragile X syndrome is caused by *FMR1* gene disruption on the X chromosome that prevents the expression of FMRP, which is involved in synaptic development and plasticity [24]. The disruption of this mechanism affects multiple neurotransmission neuromediator systems, including GABAergic. Studies have found a reduction in the expression of several GABA-A receptor subunits (namely α1,3, 4, β1,2, and γ1,2), suggesting a decrease in phasic inhibition in FXS [72]. Later experiments have reported a decrease in the number of δ subunits, which are located extra-synaptically [73] and are involved in tonic inhibition and the general level of excitability of neural networks [74]. Thus, both phasic and tonic inhibition systems are affected in FXS [75].

#### 2.1.2. Sensory Processing

Patients with FXS often demonstrate sensory hypersensitivity, which manifests as auditory, tactile, and visual defensiveness or avoidance [27,28,29]. Mice models of Fragile X syndrome also present impairments in sensory processing in visual [76], auditory [77,78], and sensorimotor modality [79], with the frequent occurrence of hypersensitive behavior, such as a higher susceptibility to seizures in response to auditory stimuli, and defensive behavior to repetitive whisker stimulation.

#### 2.1.3. EEG: Resting State

Epilepsy prevalence in FXS varies between 10% and 20%, with the mean age of onset before 10 years old [80,81]. Seizure patterns on the EEG reflect features of benign focal epilepsy of childhood and centrotemporal spikes [80,82], suggesting a shift of excitation/inhibition (E/I) balance towards excitation.

Broadband resting-state γ power (30–100 Hz), which was suggested to index the overall increase in not-well-synchronized cortical activity across local circuits [83], was elevated in FXS patients and its animal models [84,85], supporting cortical hyperexcitability in this syndrome. 

Studies of resting-state EEG in patients with FXS have also identified a decrease in α power (8–13 Hz) and stronger activity in θ band (4–7 Hz) compared to controls [86]. α rhythm (8–13 Hz) is a dominant oscillation in eye-close condition, and is proposed to reflect inhibition in sensory processing [87]; thus, its attenuation also points to an increase in the E/I balance. As Wang and colleagues [86] also observed, increased θ-γ coupling in FXS patients, along with the increased θ power being related to a decrease in resting γ power (30–80 Hz), the increase in θ power in FXS may represent a compensatory inhibition of high-frequency γ band activity, which is, however, only partially and variably successful given the observation of abnormally increased γ power in FXS.

#### 2.1.4. EEG: Evoked Responses

Despite increased resting-state γ-activity, inter-trial phase coherence of sound-evoked γ activity was reduced in response to pure tones stimulation [88]. This finding supports the recently emerging view on the different mechanism underlying resting state and stimuli-provoked γ, with the latter being linked to the feedback inhibition from PV interneurons [83]. Thus, decreased phase-locked γ activity also points to the increase in E/I balance, but this time caused by the attenuation of inhibition. Of note, this reduction in evoked γ oscillations was correlated with hyperactivity symptoms, pointing to clinical relevance [88]. Moreover, a similar γ-band reduction in response to sounds was found in the animal model of FXS, suggesting this measure as a promising translational biomarker of FXS dysfunction [84,89].

Several studies revealed enlarged N1, P2, and N2 components in response to tones presentation [90,91,92] or to white noise bursts [93], pointing to hypersensitivity in FXS. While increased N1 has been replicated consistently, N2 was also shown to be reduced in one study [88]. However, another study not only replicated the N1, P2, and N2 increase in FXS patients, but also found that N2 latency is atypically increased in these patients [94]. Attenuated habituation on the N1 component in response to repeated stimuli could cause enlarged amplitudes and was indeed found in FXS patients [88,93]. The opposite effect of the increased repetition suppression was found for phonemes in FXS patients [94], suggesting that increasing the complexity of stimuli might differentially affect the habituation process in FXS. Greater N1 amplitude was also associated with decreased phase-locked γ activity, suggesting decreased inhibition as a common underlying cause for both these abnormalities in FXS [88].

Van der Molen and colleagues [91] used an oddball paradigm, which includes a presentation of frequently occurring standard tones (1000 Hz) and rare deviant tones (1500 Hz), to study the automatic process of change detection in patients with FXS. Participants watched silent movies while the mismatch negativity (MMN) component, calculated as the difference of ERPs to standard and deviant stimuli, was measured. MMN, which occurs approximately 120 ms post-stimulus and is associated with automatic stimuli discrimination [95,96], was reduced in FXS patients. This finding corresponds to impairments of N1 habituation, reviewed in a previous paragraph. According to the “neural adaptation” hypothesis of MMN origin, the repeated presentation of standard stimuli leads to attenuated responses of feature-selective neurons. Rare sounds, in turn, activate less-adapted neurons and, thus, elicit a larger N1 response, which yields the MMN [97]. As N1 habituation is deficient in patients with FXS, the response to standard stimuli is less habituated with repetitions, causing the standard vs. deviant difference, MMN, to be less pronounced. These findings suggest a link between the impairment of the inhibition system and increased sensory responsiveness with the reduced stimulus discrimination in individuals with fragile X syndrome.

Studies in visual modality also pointed to an increased neurophysiological response at the early stages of stimuli processing: N1 and N2 amplitudes were increased in response to checkerboard and smiley faces in patients with FXS [94,98]. Rigoulot and colleagues [99] found that N170 amplitude was higher for repeated face stimuli in patients, but not in controls, again pointing to a reduction in habituation, but this time in visual modality and for more complex stimuli. 

Attentional mechanisms seem to also be affected in patients with FXS. The ERP index of automatic attention switch, positive component with a latency about 300 ms (P3a) [97], was assessed in an oddball task with the sounds presenting in the background while patients with FXS focused on silent movies. P3a was attenuated in individuals with FXS [92], pointing to the deficits in automatic attention switch. Other studies used an explicit sound discrimination task and revealed a drastic attenuation in the P3b response that reflects stimulus evaluation and decision-making processes [98,100]. Of note, early components N1 and P2 were also increased in this explicit task, similar to what was reported when the sounds are presented as a background. Thus, sensory hyperexcitability did not help to direct attention to the stimuli, but hinders its discrimination as assessed by decreased hit rates observed in patients with FXS in this task. As shown by Van der Molen and colleagues [98], the problems were not specific to auditory modality, but were also present with a visual stimuli discrimination task. These findings suggest that both the automatic detection of changes and voluntary attention switch might be impaired in subjects with fragile X syndrome.

In summary, studies of FXS consistently reflect hyperexcitation and hyperresponsiveness of the auditory system in patients and animal studies on both behavioral and electrophysiological levels, suggesting a neurophysiological pathway into the origin of these autistic symptoms. Of note, the neurophysiological hypersensitivity and decreased habituation that led to an increased response to the repetitive stimulation in the case of FXS (probably in both visual and auditory modality) might lead to decreased sensitivity to changes and inattention symptoms, as reflected in attenuated MMN and P3a components. This pattern of results might be linked to the decreased tonic and phasic inhibition found in this syndrome.

### 2.2. Angelman Syndrome

#### 2.2.1. General Info

Angelman syndrome is a genetic neurodevelopmental disorder with a prevalence rate of 1:10,000–20,000 births [101]. The clinical phenotype includes intellectual disability that ranges from moderate to severe and is present almost in 100% of patients, microcephaly, and impairments in motor development, i.e., abnormal gait and coordination difficulties and hypotonia [40,41]. Language development in individuals with AS is severely impaired, from entirely non-verbal behavior to speaking a few words or phrases [42]. Hyperactivity is present in AS, though it decreases with age [36,43]. Autistic-like characteristics are a common symptom and affect 36% of AS patients [35].

Two main genetic causes lead to AS. The first one (nondeletion type) is connected with the *UBE3A* gene de novo mutations, imprinting defects, and paternal uniparental disomy and account for 25–30% of all cases. Paternal copies of *UBE3A* are epigenetically silenced in the brain; therefore, the maternal inactivation of *UBE3A* causes a nearly complete loss of UBE3A selectively in the brain [33]. *UBE3A* encodes ubiquitin-protein ligase E3A, which is involved in targeting proteins for subsequent degradation by proteasome [102], and is also involved in the regulation of the cell cycle, synaptic plasticity, and cellular protein quality control [103]. The second cause is deletion at the 15q11-q13 locus, which contains *UBE3A* and in several cases other genes, including those coding different GABA receptors subunits—*GABRB3*, *GABRA5*, and *GABRG3* [34]. Individuals with deletion AS have a more severe clinical phenotype [41,104].

#### 2.2.2. Sensory Processing

The current criteria for a clinical diagnosis of AS includes a number of features that can be related to the sensory abnormalities, such as movement and balance impairment, hypermotoric behaviors, increased sensitivity to heat, attraction to water, and chewing and licking objects behavior [36,37,38]. Walz and Baranek [39] reported both hypo- and hyperresponsive behavior in different modalities in a large sample of AS patients. They found that hyporesponsiveness decreases with age, whereas hyperresponsiveness is more developmentally stable. Another study showed an increased level of sensory-seeking behavior that is usually connected with hyposensitivity [27]. Among anecdotal reports, an atypical response to sounds was also found, for example, to the sound of a tuning fork [105].

Animal studies also supported the hyperresponsiveness hypothesis in AS with *UBE3A* knockout mice found to be prone to audiogenic seizures [106,107]. Enhanced behavioral responses were also reported in mechanoreception and thermoreception [108]. 

#### 2.2.3. EEG: Resting State

Over 80% of patients have epilepsy, usually involving multiple seizure types and typically starting before 3 years of age [109,110]. However, EEG abnormalities can be profound even in the absence of clinical seizures, with a more severe representation in patients with the deletion phenotype [111,112,113].

The most frequently replicated EEG marker of AS is increased Δ rhythm, which is reported for both patients and mice models of the syndrome [114,115,116]. Interestingly, Δ phenotypes were more pronounced in an early age, as it was established in a longitudinal study [117], suggesting it as one of the earliest neurophysiological signs of AS. A recent study showed that the abnormal Δ-band EEG in AS is related to AS severity, underlining its significance in the pathophysiology of AS [118]. Moreover, the authors found a longitudinal correlation between EEG Δ-band abnormality and AS severity, demonstrating its potential to serve as a short-term surrogate measure for the objective assessment of treatment effects.

Animal studies have allowed us to track the Δ enhancement to the disruption in GABAergic circuits: Judson and colleagues [106] established that increased Δ rhythm and seizure susceptibility were elicited by the loss of *UBE3A* expression in GABAergic neurons. In contrast, glutamatergic *UBE3A* did not lead to EEG abnormalities. In addition, *UBE3A* function reinstatement in GABAergic neurons led to the normalization of Δ rhythm in these animals. 

Diminished β power and elevated θ power are also seen in AS patients, but only with the deletion genotype [114], suggesting the influence of the *GABRB3*, *GABRA5*, and *GABRG3* genes impairments on this phenotype.

#### 2.2.4. EEG: Evoked Responses

ERPs findings in AS are rather sparse. Key and colleagues investigated ERPs in response to words in non-verbal AS patients [118,119]. The authors found that a larger ERP amplitude of 200–500 ms component for repeated nonwords compared to novel ones were associated with caregiver reports of more adaptive communication skills. They also established that more positive ERP indices for one’s own name compared with a stranger’s name were associated with higher scores for interpersonal relationships and receptive communication. While these studies illustrate the potential of using passive paradigms for studying speech processing in non-verbal individuals, ERPs obtained in these studies lack typical ERP components and are rather noisy, thus not allowing one to infer on sensory processing in AS. 

Abnormal somatosensory ERP were reported in AS. Egawa and co-authors used magnetoencephalography (MEG) to record somatosensory-evoked fields (SEF) in response to median nerve stimulation at the wrist in AS patients with the non-deletion cause of the disease, AS patients with deletions, and control subjects [120]. The N1m peak was significantly enhanced, prolonged, and more widely dispersed in AS patients with deletions compared to non-deletion AS patients and controls, pointing to cortical hyperexcitability. The P1m was not registered in seven of eleven patients with deletions, and when it was identified, its peak latency was significantly delayed. These results might indicate a desynchronized somatosensory response in the AS deletion phenotype. SEF waveforms of AS patients with no deletions were similar to control individuals. Authors argue that impairment in GABA-mediated inhibition caused by *GABRB3* gene deletion leads to an augmentation of neural excitation and its duration. This corresponds with reports about persistent hyperresponsiveness in AS children that are mentioned above. In later studies conducted on AS mice models, this group confirmed hyperresponsiveness traits and showed that GABA-modulated tonic inhibition is decreased in *UBE3A*-knockout mice [121].

In addition, Guerrini with colleagues [122] used transcranial magnetic stimulation to elicit motor evoked potentials and showed that the silent period following motor evoked potentials was shortened by 70%, indicating motor cortex hyperexcitability.

In summary, even among the patients with similar genetic etiology and relatively homogeneous symptoms that lead to diagnosis of AS, we can see heterogeneity in the sensory processing with both hypo- and hyperresponsiveness reported at the behavioral level. This heterogeneity might be partly caused by the number of genetic abnormalities involved and the implication of additional genes (such as those related to GABA activity) that should be taken into consideration. On the electrophysiological level, we can see hypersensitive patterns, such as frequently occurring seizures, prolonged components of somatosensory ERPs, and generally decreased neuronal noise that can be suggested from elevated slower rhythms (θ and Δ) over the higher ones (β) [123] reported for a subpopulation of AS with the deletion phenotype. Combining these findings with animal models, we can assume that the impairment in tonic inhibition leads to the hypersensitive phenotype in patients with AS.

## 3. Hyposensitive Syndromes

### 3.1. Phelan–McDermid Syndrome

#### 3.1.1. General Info

22q13 deletion syndrome, also known as Phelan–McDermid syndrome (PMS), is a neurodevelopmental disorder characterized by hypotonia, intellectual disability of varying degrees (from severe in a majority of cases to mild in some patients), delayed or absent expressive speech, and dysmorphia [45,50]. Its prevalence according to approximate estimates is between 1:8000–15,000 and the rate of autism in PMS is very high and varies from 50 to 75% [45,46]. It is supposed that 1% of people with autism have Phelan–McDermid Syndrome [124].

PMS is caused by de novo or inherited impairments at the 22q13 locus with the *SHANK3* gene as the major candidate gene. The severity of symptoms positively correlates with deletion sizes [44]. The protein product of *SHANK3* (it has the same name) is a scaffolding protein in postsynaptic glutamate receptors, including NMDA, mGluRs, and AMPA receptors [125,126]. *SHANK3* knockout mice demonstrate a reduction in the expression of parvalbumin (PV), while the amount of GABAergic PV+ interneurons does not change [127]. As decreases in PV leads to increased facilitation of GABA release, the inhibition is increased [128]. A series of animal studies demonstrated that *SHANK3* deletion only in the GABAergic interneurons is enough to reproduce the phenotype [129,130], supporting the crucial role of inhibition in the development of PMS.

#### 3.1.2. Sensory Processing

Mieses and coauthors reported that 80% of patients with PMS had sensory abnormalities in tactile, auditory, and visual modalities; however, the most severe impairments were in the low energy domain that refers to motor hypotonus [47]. Another study exploring the phenotype of the syndrome found that 80% of the individuals had a high sensory threshold [48]. In a more recent study, children with PMS demonstrated significantly greater hyporeactivity symptoms compared to children with idiopathic ASD and the control group across all modalities [49]. Authors divided patients into two groups based on the size of the mutation: the first group included children with only the *SHANK3* gene affected and the second one included the children with larger deletions. No differences were found in those two groups, leading to the suggestion that disruption of the *SHANK3* gene is enough for sensory impairments. Thus, PMS can be considered as a hyposensitive syndrome based on clinical and behavioral evidence in patients.

Patients’ data is also supported by animal studies, where hyposensitivity in somatosensory and auditory modality was reported in various types of *SHANK3* disruption models [131,132]. However, in another study, the *SHANK3* gene was selectively deleted in inhibitory interneurons and this has led to the increased activity of pyramidal neurons and tactile hypersensitivity in mice [129]. That was different from the complete disruption of *SHANK3* and the occurrence of hyposensitivity features in this case [132]. This discrepancy is hard to account for right now and might be a good topic for further research.

#### 3.1.3. EEG: Resting State

Patients with PMS have a higher risk of developing seizures with a reported prevalence 40–63% [133,134], and an even higher presence of epileptiform and disorganized activity, with a general slowing of background EEG [135,136]. Spectral EEG changes in PMS include generalized slowing of activity, reduced occipital α rhythm [136,137,138,139], and decreased β and γ rhythm [140].

#### 3.1.4. EEG: Evoked Responses

ERP studies in PMS show a reduction of ERP components in various modalities, but the most studied is the auditory one. 

Auditory steady-state response (ASSR) to 40 Hz stimulation is a reliable way to assess γ-band oscillation driven by repetitive stimulation, e.g., click trains. One study reported weaker γ-band ASSR in patients with PMS as well as with idiopathic ASD [141]. However, data on 40-Hz ASSR in idiopathic ASD is inconsistent as no difference in this response was also reported [142,143], suggesting that atypical γ-band ASSR might be associated with a subgroup of ASD, probably those with abnormalities within a *SHANK3*-related pathway. This idea has been supported by a recent case study of a high-functioning girl with mild intellectual disability, attenuated language skills, and slightly atypical social and communication functions. She had a drastic reduction of 40-Hz ASSR and *SHANK3* microduplication [144], again linking γ-band ASSR with *SHANK3* abnormalities. As 40-Hz ASSR reflects the functioning of NMDA receptors on GABAergic interneurons [145,146], its reduction supports the link between the abnormal functioning of local inhibitory connections in PMS patients and potentially in a subgroup of idiopathic ASD. 

PMS individuals also showed a decreased amplitude of the P50 component to the presentation of repeated tones [147] and to broadband noise bursts [148], as well as a reduction in P2 amplitude and its stronger habituation, pointing to possible impairments in plasticity mechanisms [147]. Ponson and colleagues conducted a study in patients with PMS using an oddball paradigm and showed that patients meeting criteria for autism have unaffected MMN, but tended to have longer latencies of the N250 component in response to tones as compared to controls [149]. This component is known to reflect speech structure processing and has been related to language abilities in children [150]; thus, delayed N250 reported in PMS patients with ASD might indicate longer auditory processing time relevant to language functions.

Similar electrophysiological properties were shown in animal models. Engineer and coauthors documented the auditory cortex response to speech and non-speech sounds in the *SHANK3*-deficient rat model [151]. The number of spikes evoked by tones was significantly smaller in *SHANK3* heterozygous rats compared to control rats, as well as the spontaneous firing rate. The consistent pattern was seen in response to rapid trains of sound and speech stimuli. Authors have concluded that auditory responses generally are degraded in *SHANK3*-deficient rats due to neural hyporesponsiveness. 

In the visual modality, the amplitudes of P60-N75 and N75-P100 were also decreased [140,152], pointing to hyposensitivity of the visual cortex. The P60-N75 amplitude was negatively correlated with deletion size, providing evidence for the clinical relevance of this measure [140].

We can see that hyposensitivity on a behavioral level corresponds to findings of hypoexcitability in both patient and animal experiments. For example, ERP components are decreased in both visual and auditory modalities. Animal studies show the involvement of the GABAergic system, particularly of PV+ interneurons, into the development of this phenotype; however, future research is needed to understand this mechanism more precisely.

### 3.2. Rett Syndrome

#### 3.2.1. General Info

Rett syndrome (RS) is a severe neurodevelopmental disorder generally characterized by normal development during the first 6–18 months, followed by a stagnation and a loss of acquired motor and language skills. RS is frequently associated with autistic traits which occur in approximately 60% of cases [35]. Other common symptoms are breath irregularities, abnormal gait, and hand wringing [57,153]. Its prevalence varies from 1:10,000 to 1:20,000 [58]. RS patients are suggested to have severe intellectual disabilities [154], while adequate assessment of cognitive functions are challenging due to severe motor problems and the absence of necessary language skills.

The majority of Rett syndrome cases come from different types of *MECP2* gene mutations. Its protein product—*MECP2* protein—interacts with a regressor complex of HDACs and SIN3A proteins to repress gene transcription [51,56] and also plays the role of a transcriptional activator [52]. *MECP2* affects the activity of more than 60 different molecular pathways, including those involved in spine morphology, dendritic complexity, and mTOR signaling vital for cell growth and metabolism [155]. Of note, *MECP2* disruption only in GABAergic neurons seemed to be enough to cause symptoms in mice models [156], pointing to the crucial role of inhibition in the development of RS.

#### 3.2.2. Sensory Processing

Sensory processing is reported to be impaired in patients with Rett syndrome, but data is sporadic and inconsistent. Pain sensitivity is the most frequently described modality and is decreased in patients. Mice with a knockout of *Mecp2* revealed hypersensitivity to pressure and cold and hyposensitivity to heat [157]. Thus, from the behavioral data, it is not clear whether and how sensitivity is impaired in different modalities.

#### 3.2.3. EEG: Resting State

Epilepsy is quite common in Rett syndrome and reported in up to 70% of patients, with an even higher incidence of epileptiform activity [55,158]. The most frequently reported types of epileptiform activity are centrotemporal spikes, which are also common in fragile X syndrome as mentioned above.

Besides that, background EEG in patients with Rett syndrome has a tendency to slow down and it is observed even at the earliest stage of the disorder (for a review of resting-state EEG as a translational marker of Rett syndrome, see [159]). In particular, attenuated α and β band power, with an accompanying increase of θ and Δ activity, were reported. Furthermore, enhanced Δ activity was correlated with the severity of cognitive impairments [160], pointing to the clinical relevance of this neurophysiological abnormality. 

As for the high-frequency domain, animal studies revealed a decrease in γ-band frequency in knockout models of the syndrome [161,162].

#### 3.2.4. EEG: Evoked Responses

One of the most sustained results of ERPs studies of Rett syndrome is an increased latency of the ERP components that is observed in all modalities and coincide in patients and animal model studies (review in [163]). 

For the auditory modality, abnormalities are more pronounced for later ERP components, suggesting that an auditory deficit occurs at higher levels of sensory processing. For instance, it was shown that P2 and N2 components were smaller than TD’s value in almost all RS patients in response to tones and phonemes, whereas P1 and N1 did not differ between groups [164]. At the same time, in one study N1 and P2 had longer latencies [165]; however, this finding was only replicated for P2 [164]. Moreover, P2 amplitude was correlated with RS severity and similar changes in later auditory components were observed in animal models of RS [166], suggesting that they are promising translational biomarkers of RS severity.

MMN for change in the frequency of auditory stimuli was also delayed and increased in duration in patients with RS, suggesting a deficit in automatic change detection and slowing of auditory responsiveness [167]. In another study conducted by Brima and colleagues [168], MMN for changes in stimuli duration was decreased with short interstimulus intervals ((450 ms) and absent with longer intervals (900 and 1800 ms) in patients with RS. This can indicate that, even if patients with Rett syndrome can indicate changes in stimulation at a fast rate, this ability is altered for the slower speed of presentation.

In the visual modality, the early components of ERPs are also impaired (review in [163]). LeBlanc and coauthors showed that a decreased amplitude of visual-event-related potentials can be a RS translational marker [169]. In this study, the checkerboard pattern was presented to patients with Rett syndromes and *Mecp2* heterozygous female mice. The amplitude of the N1–P2 complex was decreased and N2 latency was increased in patients, and mice showed a similar impairment. A delay in the P1 component was also observed in other studies [170].

Interestingly, somatosensory ERPs (the N20-P30 or P30-N35) were increased in RS [171,172], suggesting hypersensitivity that corresponds to animal studies, where hyperresponsiveness to pressure was observed [157].

Electrophysiological and phenotypical data concerning sensory deficit in Rett syndrome is inconsistent, suggesting discrepancies between modalities. ERPs studies of auditory and visual modalities suggested hyposensitivity (delayed and decreased amplitudes of P2 and MMN in the auditory modality and reduced N1-P2 and N2 components in the visual modality), whereas somatosensory studies pointed to hypersensitivity. Attenuated and delayed ERPs, as well as steeper 1/f slope of resting EEG with predominance of low-frequency over the high-frequency activity, might be linked to increased tonic inhibition suggested in RS. One possible mechanism of this imbalance in RS is based on increased GABA spillover beyond the synapse [159]. The study conducted by Zhang and colleagues provided the support for the causative role of tonic inhibition for seizure generation in mice models of RS [173]. Authors showed that inhibition of the GAT-1 protein, which is involved in GABA reuptake, can result in increased cortical epileptiform discharges. Even if we cannot directly apply this finding to the development of sensory deficit in RS, this study proposes that increased tonic inhibition might be a possible pathway of sensory abnormalities as well.

### 3.3. Tuberous Sclerosis

#### 3.3.1. General Info

Tuberous sclerosis complex (TSC) is a genetic disorder with the prevalence of 1:5,500–10,000 live births in the population. It is characterized by benign tumors in multiple organs, epilepsy, attention deficit and hyperactivity disorder, anxiety, sleep disorders, and other behavioral problems [63]. Severe intellectual disability is present in more than half of patients [62]. ASD criteria meet around 25–40% of patients [60,61].

TSC is caused by a mutation in the *TSC1* (encodes hamaritine protein) or the *TSC2* (encodes tuberine protein) genes. Tuberine and hamaritine act as a complex and inhibit the RHEB protein, which activates mTORC1, a crucial protein for many neurodevelopmental processes, such as the initiation of protein translation and cell growth control [59].

#### 3.3.2. Sensory Processing

Little is known about hypo- or hypersensitivity tendencies in sensory processing in TSC patients; however, in a case study, sensory dysregulation in daily activities was reported [174]. Animal models’ investigation does not clarify the question. Sato and coauthors reported that *TSC2* knockout mice had no differences in sensory processing compared to wild-type mice [175]. Taking this into account, it is even more interesting to see if a deficit in sensory processing can be seen on an electrophysiological level.

#### 3.3.3. EEG: Resting State

Epilepsy is quite common in TSC, affecting 80–90% of patients, and has an early onset: 6% of patients develop seizures during the first month of life and 60–70% develop epilepsy during the first year [176]. Another study revealed that a significantly greater amount of interictal epileptiform features in the left temporal lobe were associated with the presence of ASD symptoms in TSC patients, suggesting the importance of this brain region dysfunction for ASD development [60]. 

A quantitative analysis of resting-state EEG rhythms has been limited in TBS to early childhood (up to 2 years old). Dickinson and colleagues reported no difference in α power during attentive wakefulness [177]. However, the α peak was shifted to lower frequencies in TCS at age 24, but not 12, months. This finding might be related to delayed maturation of the thalamocortical path in TSC. α phase coherence was, on the other hand, atypical at age 12 months, but not 24 months. Of note, similarly decreased α coherence between left temporal and right central regions differentiated TCS patients with and without ASD at 24 months. Thus, altered long-range interhemispheric connections characterize TCS patients from an early age and might indicate delayed maturation of white matter during infancy that later led to the development of ASD symptoms. In addition, α peak frequency at 24 months predicted cognitive functioning at 36 months across all infants. Another study conducted on a different sample confirmed the predictive value of EEG abnormalities in TSC patients. De Ridder and colleagues reported that a dysmature EEG background and increased Δ activity is associated with a higher probability of ASD traits at the age of 24 months [178]. Thus, connections between early EEG patterns and later development of cognitive functions highlight the possibility of using early EEG markers to identify infants with TSC who need additional targeted intervention.

#### 3.3.4. EEG: Evoked Responses

ERP studies point to sensory processing being impaired in TSC patients in different modalities. 

In the auditory modality, the low amplitude of the P1–N1 complex in response to three consecutive presentations of phoneme was found in TSC patients [95]. While this study did not segregate patients in relation to ASD diagnosis, other studies reported significant differences within TSC patients related to ASD symptomatology. Seri and coauthors found an impairment in auditory ERPs in those TSC patients who were diagnosed with autism: N1 had a significantly smaller amplitude and prolonged latency and mismatch negativity (MMN) had a longer latency [179]. Importantly, magnetic-resonance imaging also revealed lesions in the temporal lobe in a subgroup of TSC patients with ASD, pointing to those structural abnormalities possibly underlying the altered auditory processing in this patient group. In another study, researchers presented vowels and non-speech stimuli to TSC patients and found that typically developed children had increased P1 and N2 amplitudes to vowels compared to tones, whereas patients with TSC did not [180]. Interestingly, there was no difference in MMN for both vowels and tones between TSC and control groups; however, in children who had TSC and ASD, MMN was significantly lower than in TSC children without ASD and the control typically developing subjects. These findings reflect a deficit of auditory processing at the early stages and also point to an impairment in automatic changes detection that corresponds to the presence of the autistic symptoms.

In the visual modality, longer latency of the N170 and N290 component in patients with TSC was demonstrated compared to typically developed children during the presentation of human faces [181,182]. P1 latency was also affected during face perception: control subjects had longer latency to inverted faces, whereas TSC patients did not [182], indicating impairments even in earlier steps of visual processing. In these studies, participants were adolescents or adults, and it is interesting that in infants with tuberous sclerosis, no differences in visual ERPs to black-and-white checkerboard were found compared to typically developed infants [183]. Thus, abnormalities in visual processing did not manifest itself from birth but might develop later with a progression of the disease or be specific to social stimuli. 

In summary, electrophysiological evidence suggested hyposensitivity in patients with TSC (delayed visual components and decreased and delayed ERPs in auditory modality). Interestingly, despite downregulated inhibitory currents [184], this syndrome remains mostly hyposensitive. One possible explanation for this was provided from a magnetic resonance spectroscopy (MRS) study, where a decreased number of GABA-A receptors was found in patients with TSC, accompanied by elevated GABA concentration [185]. Authors suggested the elevation of GABA is a compensation mechanism for decreased inhibition. Possibly, the molecular impairments associated with TSC lead to overcompensation and result in a hyposensitive phenotype.

### 3.4. Neurofibromatosis Type 1

#### 3.4.1. General Info

Neurofibromatosis type 1 (*NF1*) is the most common inherited disorder of the central nervous system with a prevalence that varies from 1:2,000 to 1:5,000 subjects [186]. Phenotypically, *NF1* is characterized by multiple cafe-au-lait spots (flat dark patches on the skin), axillary and inguinal freckling, and various and multiple benign tumors in the central nervous system. Patients with neurofibromatosis type 1 are likely to have ASD and its prevalence varies from 10 to 40% [65,66]. Moreover, ADHD is very common in this disorder, being a major contributor to academic underachievement in this population [70]. Intellectual disabilities are not very common in *NF1*; however, it occurs in a mild degree in 5–30% of patients [67,68].

*NF1* is caused by mutations in the *NF1* gene located at chromosome 17q11.2; they can be either inherited or de novo [64]. This gene encodes a neurofibromin protein, reduces cell proliferation, and is involved in the development and growth of a variety of tissues [187]. An animal model of *NF1* is characterized by an increased activity-dependent release of GABA, suggesting increased phasic inhibition in *NF1* [188].

#### 3.4.2. Sensory Processing

The sensory deficit is common in patients with *NF1*, reported even in infants with this disorder [189]. The main features of sensory processing included a low threshold for noticing sensory changes and reduced attention towards environmental cues in visual, tactile, and auditory modality. Whether these problems are due to attentional problems or pure sensory abnormalities are in place are challenging to track with behavioral methods. Electrophysiological data might help to disentangle these possibilities.

#### 3.4.3. EEG: Resting State

Neurofibromatosis type 1 is associated with epileptic seizures in 4–7% of patients [190]. Resting-state EEG is characterized by elevated θ and α power in *NF1* patients, while changes in α power was only marginally significant [191]. Other frequency bands were unaltered in *NF1*. The functional significance of these findings is unclear.

#### 3.4.4. EEG: Evoked Responses

Complications in *NF1* include deafness and optic gliomas that are related to vision loss or eye-bulging [192,193]. Thus, auditory and vision functions are regularly monitored in *NF1* patients.

Increased latencies of brainstem auditory evoked potential (bAEP) were found in 15–28% of *NF1* patients [194,195]. Findings from a very recent study suggest that abnormalities in auditory processing in the early stages of development can cause the occurrence of ASD symptoms in *NF1* [196]. Authors investigated ERPs to repetitive auditory stimuli in infants with *NF1* at the age of 5 and 10 months and found decreased age-related changes in the scalp profile of neural responses. Moreover, the pitch-change detection response was delayed in infants with *NF1*. These ERP abnormalities correlated with early symptoms of ASD behavior measured at 14 months.

Abnormal visual ERP was reported in about half of *NF1* patients that showed a delay and decrease in amplitude of the P100 component [192,194,195,197,198].

In addition to ERPs studies, α power is also impaired in *NF1* patients during visual tasks: Ribeiro and co-authors [191] showed an enhancement of both phase-locked and non-phase-locked α oscillations during visual stimulation. At the same time, functionality of α is preserved in *NF1* as it is even higher in closed-eyes condition. Elevated α can reflect difficulties in visual discrimination ability [199,200] as well as in attentional suppression mechanisms when visual stimuli need to be ignored [201].

Impairment in somatosensory ERPs are also present in a substantial proportion of *NF1* patients ranging from 15–35% that showed delayed components and central conduction time [194,195].

Besides perceptual processing, attention and executive function deficit is in the spotlight of EEG research of *NF1*. Many patients have symptoms of attention deficit disorder with or without hyperactivity, and its variety is between 30 and 50% [202]. Ribeiro and coauthors studied ERPs in the Go-No Go paradigm (subjects were asked to respond to all digits, but ‘3’) [203]. Behaviorally, patients with *NF1* failed to inhibit the button press in the No-Go trial. At a neurophysiological level, P1 amplitude was significantly reduced in patients, suggesting that attention problems might be due to attenuated low-level sensory sensitivity. Later positive response over centro-parietal scalp sites elicited for No-Go stimuli was preserved in *NF1* patients. In contrast, frontal P3, which has been associated with go/no-go and stop signal tasks performance [204], was significantly reduced in the *NF1* group. Using magnetic resonance spectroscopy (MRS), they found a reduced level of GABA in the posterior prefrontal region and the occipital cortex. This finding was replicated by a positron-emission tomography (PET) study, where authors showed reduced GABAA receptor density in *NF1* patients [205]. Thus, reduced inhibition might trigger inattentiveness manifested in frontal P3 attenuation. Reduction in P1 can be the sign of a different mechanism and more compatible with a notion of increased inhibition and hyposensitivity. Indeed, the amplitude of a frontal P3 neuroelectric signal was not correlated with P1, suggesting independent neuronal mechanisms.

Bluschke and colleagues conducted a study of response inhibition in adolescents with *NF1* using a flanker test, where subjects were asked to press the right/left button if the arrow in a screen point to the right/left, respectively [206]. Two distractor arrows, pointed either to the same direction (67% of trials) or to the opposite, were presented 200 ms before the target. Behaviorally, patients with *NF1* perform significantly slower, but more accurately, than controls. At the neurophysiological level, the occipito-parietal P1 and N1 components following the flanker, as well as the target stimuli, were unremarkable in *NF1* patients, pointing to undisturbed early stages of sensory processing. However, N2 and N450 components that have been associated with conflict monitoring were atypical. In particular, the N2 component was reduced for incompatible trials in *NF1* compared to controls, while N450, on the other hand, was enhanced, but predominantly for compatible trials, as in controls, this response was sensitive to conflict and decreased where no conflict was detected. *NF1* patients lack this differentiation in N450, which was similarly high for both compatible and incompatible trials. Authors presumed that *NF1* patients maintain a higher level of cognitive control resources to compensate for the atypical attenuation of response at the preceding stage of stimuli processing.

Pobric and colleagues studied *NF1* patients’ performance and ERPs during a working memory task [207]. They reported that *NF1* patients have a quicker P3 component, which is usually associated with attention allocation and memory updating. A decrease in P3 latency can be interpreted as a quicker information processing time [208]. Taken together with the lower accuracy in task completion seen in these patients, the ERP finding suggests faster, but less effective, neurophysiological processes subserving the performance in this complex cognitive task.

Thus, in addition to delayed and decreased early components in different modalities, especially in the visual modality, research focuses on impaired attention in neurofibromatosis type 1. A deficit of attention-related component P3 was reported and the possible cause for that is a reduced sensitivity to sensory stimuli that significantly impact the bottom-up attention switch; however, it failed to be proven experimentally. Interestingly, a later component N450 was shown to be enhanced, which probably corresponds with a compensation mechanism. Taken together, the increased phasic inhibition might be related to attenuated early ERP components, while lower inhibition in the frontal cortex might cause the quicker and smaller attention-related ERPs.

**Table 2 genes-13-00671-t002:** Summary of resting-state EEG and ERP findings in different syndromes: Fragile X syndrome (FXS), Angelman syndrome (AS), Phelan–McDermid syndrome (PMS), Rett syndrome (RS), tuberous sclerosis (TSC), and neurofibromatosis type 1 (*NF1*). ↑/↓ means increase/decrease respectively.

Syndrome	Epilepsy, Onset	Resting-State EEG	Auditory EEG	Visual EEG	Somatosensory EEG
FXS	10–20%, before 10 y.o. [80,81]	↑ γ power [84,85] ↓ α power [101] ↑ θ power [86]	↓ evoked γ power [88] ↑ N1, P2 amplitude [90,91,92,93] ↓ MMN, P3a, P3b amplitude [92,99] ↑↓ N2 amplitude [89] ↑↓ N1 habituation [88,92,94] ↑ N2 latency [54]	↑ N1, N170, N2 amplitude [94,98,99] ↓ P3b amplitude [98] ↓ N170 habituation [99]	n/a
AS	80%, before 3 y.o. [109,110]	↑ Δ power [114,115,116,117] in patients with deletion: ↓ β power [114] ↑ θ power [114]	↑ amplitude 200–500 ms for repeated non-words was associated with better communication skills [118,119]	n/a	in patients with deletion: ↑ N1m amplitude [120] ↓ P1m amplitude [120] Prolonged N1m, P1m component [120]
PMS	63%, between 2–6 y.o. [133,134]	↓ γ power [140] ↓ β power [140] ↓ α power [135,138,139]	↓ 40Hz ASSR [142] ↓ P50, P2 amplitude [148,149] ↑ P2 habituation [148] ↑ N250 latency [150]	↓ P60-N75, N75-P100 amplitude [140]	n/a
RS	60–80%, between 2–4 y.o. [158]	↓ β power [159] ↓ α power [159] ↑ θ power [159] ↑ Δ power [159,160]	↓P2 and N2 amplitude [164] ↓ MMN amplitude [167,168] ↑ N1 and P2 latencies [164,165] ↑ MMN latency [167]	↓ N1-P2 amplitude [169] ↑ P1, N2 latency [169,170]	↑ amplitude of SEP [171,172]
TSC	80–90%, before 2 y.o. [176]	↓ α peak frequency [177] in TSC with ASD: ↑ Δ power [178]	↓ P1-N1 amplitude [95,179] in TSC with ASD: ↓ MMN amplitude [179] ↑ N1, MMN latencies [179]	↑ N170 and N290 latency in response to human faces [181,182]	n/a
*NF1*	4–7%, varies a lot [191]	↑ θ power [191] ↑ α power [191]	↑ latency in brainstem auditory evoked potential [194,195] ↑ latency in pitch-change detection response in infants [195]	↑ α oscillations during visual stimulation [191] ↓ P1, P3 amplitude [194,198] ↓ N2 amplitude in incompatible trials [206] ↑ N450 amplitude in compatible trials [206] ↑ P1 latency [194,195,198] ↓ P3 latency in n-back task [204]	↑ latency in SEP [194,195]

## 4. Discussion

In this review, we combined available electrophysiological studies in several syndromes associated with autism and tried to explain the behavioral phenotype through the alteration of E/I balance. Table 2 contains a summary of resting-state EEG and ERP findings in these syndromes. Based on these findings, we subdivided syndromes into two groups with hyper- and hyposensitivity phenotypes.

In Fragile X syndrome and Angelman syndrome, the core deficit seems to be hypersensitivity, which is supported by such evidence as increased amplitude and prolonged length of ERP components in different modalities. In FXS, the auditory system is the most affected, with the most consistent finding being an increased N1 amplitude possible resulting from the diminished habituation of the N1 component. Later components (N2, P2) also seem to be enhanced; however, more studies are needed to confidently draw any conclusions. In Angelman syndrome, the tactile system is more widely examined and the N1m SEF component has been reported to be enhanced, prolonged, and widely dispersed, suggesting hypersensitivity. Importantly, animal studies also support hypersensitivity in these syndromes, e.g., *FMR1*- and *UBE3A*-deficient mice are prone to audiogenic seizures.

Phelan–McDermid syndrome, Rett syndrome, Tuberous sclerosis, and Neurofibromatosis type 1 show, in general, a decreased amplitude and delayed responses related to early-stage processing. Auditory perception is the most examined area in these syndromes and its impairments have commonalities across syndromes: a decrease of P2 amplitude is seen in PMS and RS and longer latencies of N1 and MMN occur in RS and TSC. Moreover, P50 is decreased in PMS. In the visual modality, hyposensitive syndromes have decreased early components, namely P1 in PMS and *NF1* and N1–P2 in RS. Moreover, delayed P1 is a common trait for RS and *NF1*.

Thus, event-related potentials provide multi-stage information about these syndromic forms of ASD and detect tendencies towards hypo- or hypersensitivity.

### 4.1. Oscillatory Activity

We also reviewed findings on brain oscillation in rest and with stimulation, as they shed light on tendencies towards hypo- or hyper neural excitability in different syndromes. Among consistent findings across syndromes was the relative increase in the low-frequency activity (θ, Δ) and decrease in the high frequency activity (β, γ) that is rather nonspecific and reported in many neurological and psychiatric conditions, including ASD [209], ADHD [210], and Down syndrome [211]. In neurodevelopmental disorders, such a shift of dominant activity towards low frequencies can be related to developmental lag, as low-oscillation predominance characterizes the child brain [212]. Recently, this shift, that can be estimated by 1/f slope, was related to E/I imbalance [123]. When 1/f slope becomes steeper, which is the case in most of our syndromes, it points to more synchronous activity of the neuronal population, indicating a decrease in the E/I balance. However, the degree of this steepness was not directly assessed in all but RS and FXS of our syndromes. In FXS, a hypersensitive syndrome according to our scheme, the slope was flatter (representing the increase in high-band frequencies and neural noise) [213]; however, in RS, a hyposensitive syndrome, the 1/f slope was steeper (or more negative) [160]. Future research should compare 1/f in different syndromes associated with ASD as it is a promising measure of neuronal dynamics.

α oscillations were also altered in syndromic autism. In FXS, PMS, and RS, α is diminished (see Table 2), while in patients with *NF1*, both phase-locked and non-phase locked activity is elevated during visual stimulation. As α activation is considered as a sign of inhibition as its decrease points to increased E/I balance, while an increase, on the other hand, points to increased inhibition. Thus, α data obtained from FXS and *NF1* corresponds to our view on these syndromes. Findings on PMS and RS suggest a more complex relationship between α activity, E/I balance, and the neurodevelopmental phenotype. For example, a recent study [139] did not replicate lower α for PMS, but instead found alternation in α–γ phase–amplitude coupling that might be related to disbalance between top-down and button-up processing.

γ rhythm is also an interesting object of research, as it is a direct measurement of cortical excitation–inhibition interaction [214]. Currently, it has been proposed that spontaneous broadband γ power may reflect the overall level of circuit activity, whereas γ activity that occurs in a narrower frequency band in response to sensory stimulation specifically indicates feedback inhibition from PV+ interneurons [83]. In FXS, a disorder with the most consistent hypersensitivity phenotype, resting state γ activity was elevated, while in hyposensitive syndrome, in PMS, it was decreased. Evoked γ, on the other side, was downregulated in patients with FXS as well as in PMS, despite them showing an opposite pattern of sensitivity. In summary, resting state and evoked γ may reflect different aspects of inhibition and should be studied in conjunction to provide a better understanding of the E/I imbalance in different forms of autism.

### 4.2. Approach for Idiopathic ASD Stratification

Our review provides a first stretch of specific neurophysiological profiles of six syndromes associated with autism, pointing to their commonalities as well as their differences. As one of our goals is to provide an approach for subdivision of idiopathic ASD, here we compare findings in genetic syndromes with results from idiopathic ASD studies to illustrate our approach. Taking into account the behavioral heterogeneity of idiopathic ASD, it is not very surprising to see inconsistent findings at the neurophysiological level as well. For example, resting γ activity in ASD was found to be both elevated [215] as well as reduced [216]. Our review points out that some syndromes are characterized by increased spontaneous γ (FXS), while in others, it is reduced (PMS). ERP changes in idiopathic ASD, while being more consistent with what is reported for hyposensitive syndromes, which showed a decrease and delay of the major ERP components (for auditory modality see recent meta-analysis pointed to delay P1 and reduced amplitude of N1 and N2 [217], for visual modality [218,219]). However, enhancement in ERP components have also been reported in idiopathic ASD, pointing to the heterogeneity (e.g., increase P1 and N1 amplitude in auditory modality [220,221,222]). The predominance of ERP studies that report decreased amplitude and delay latency of the early ERP components in idiopathic ASD might be related to the EEG study participation bias: individuals with hypersensitivity are much more challenging to get to wear the EEG cap and less likely tolerate an experimental procedure with external stimulation than hyposensitive individuals. When the group is heterogeneous and deviates from the mean in both directions, it is hard to track the difference with a traditional statistical approach. Normative modeling, pattern classification, and stratifications might help to deal with this problem in the future. Knowledge about ERP changes in the syndromic form of ASD will allow us to link the subgroup of idiopathic ASD with a particular neurophysiological profile to underlying molecular-genetic changes. Based on our analysis, we can suggest that there is a subgroup of idiopathic ASD that shows a hypersensitive behavioral phenotype, as well as being characterized by increased γ band activity and early ERP components. Abnormalities in this ASD subgroup might be linked to the molecular-genetic paths disturbed in FXS. 

### 4.3. A possible Role of Inhibition in Sensory Abnormalities

This study tried to link abnormalities in different genes, such as *FMR1, UBE3A, GABRB3, GABRA5, GABRG3, SHANK3, MECP2, TSC1, TSC2*, and *NF1*, that are associated with synaptic transmission, cell growth, as well as translational and transcriptional regulation, with behavioral and neurophysiological sensitivity, as these genes are causative to syndromes with autism-like symptoms and sensory abnormalities. Our review suggests that the *FMR1, GABRB3, GABRA5,* and *GABRG3* genes are related with hypersensitivity while the *SHANK3, MECP2, TSC1, TSC2*, and *NF1* genes contribute to hyposensitivity at neurophysiological and/or behavioral levels. This relation might be linked with the interplay of tonic and phasic inhibition in different brain regions. 

Based on the hypothesis of excitation/inhibition balance as a possible cause of autistic syndromes, we followed the variants of the GABA system disruption in syndromic forms of ASD. Inhibitory circuits were disrupted in each syndrome. GABAergic inhibition can be phasic (synchronous opening of channels which are clustered at the synaptic junction) and tonic (activation of receptors which are distant from neurotransmitter release). Interestingly, the phasic and tonic inhibition seemed to be differently affected. For example, in Rett syndrome, decreased phasic and increased tonic inhibition was suggested [159]. Increased tonic inhibition was also reported for Angelman syndrome [121]. On the other hand, in Fragile X syndrome, both types of GABAergic inhibition were diminished [75]. Thus, disruption of tonic and phasic inhibition in different combinations can contribute to heterogeneity in sensitivity and neural excitability.

### 4.4. Concluding Remarks/Open Questions

Our main motivation for this review was to summarize the electrophysiological findings for syndromic forms of autism and relate them with sensory abnormalities that are common in these syndromes. We managed to identify several indicators that would be useful for assessing sensory impairments in both research and clinical practice. First, it is components of event-related potentials that might point to shifts towards hypo- or hypersensitivity at different stages of information processing. Second, resting state EEG can provide information about tendencies towards inhibition or excitation in the neuronal network, e.g., via such promising parameters as 1/f slope, and evoked and induced γ and α rhythms. The above-mentioned EEG/ERP parameters can be a good candidate for translational biomarkers between animal and patient studies that allow one to capitalize on the more precise understanding of mechanisms and the underlying molecular paths to sensory deficit in syndromic forms of ASD that is possible only in animal studies.

Our review also revealed gaps and inconsistencies among the studies of different syndromes. Thus, the field will benefit from a common protocol for electrophysiological and event-related potential research that should include investigation in all modalities and stages of sensory processing, as well as potential biomarkers of particular molecular changes (such as 40-Hz ASSR for GABAergic signaling). As most genetic syndromes are very rare, we welcome multi-site collaboration in this endeavor.

## Figures and Tables

**Table 1 genes-13-00671-t001:** Phenotypic description of syndromic forms of ASD: Angelman syndrome (AS), Fragile X syndrome (FXS), Phelan–McDermid syndrome (PMS), Rett syndrome (RS), Tuberous sclerosis (TSC), and Neurofibromatosis type 1 (*NF1*).

	Genetic Cause	Autism	Sensory Deficit	Intellectual Disability (ID)	Language Problem	Other Deficits
FXS	*FMR1* in chromosome X [24]	22% [25,26]	auditory, tactile, and visual defensiveness or avoidance [27,28,29]	ID varies from mild to moderate degree [25]	expressive and receptive skills are mildly delayed [30]	hypotonia, ADHD 60–80% [31,32]
AS	*UBE3A* (+*GABRB3, GABRA5, GABRG3*) in chromosome 15 [33,34]	25–40% [35]	hypo and hyperresponsiveness, hyporesponsiveness decreases with age; heat sensitivity, water attraction, sensory seeking [27,36,37,38,39]	100%, severity depends on types of mutation (moderate to severe ID [40,41]	80% delay in expressive language with minimal or no use of words [42]	hypotonia, attention deficit [36,40,41,43]
PMS	*SHANK3* in chromosome 22 [44]	50–75% [45,46]	high sensory threshold, hyporeactivity [47,48,49]	53% profound ID, 23% severe ID, 10% moderate ID, 10% mild ID [45]	100% delay, 50% had no expressive speech [45,50]	hypotonia, bipolar disorder—54% psychosis—12% irritability—36% [45,50]
RS	*MECP2* in chromosome X [51,52]	60% [35]	reduced pain threshold [53,54,55]	100% severe ID [56]	80–90% language in regression period, could say few or no words [56]	hypotonia motor stereotypes [57,58]
TSC	*TSC1* and *TSC2* in chromosome 9 [59]	36% [60,61]	n/a	64% severe ID [62]	24–66% abnormal language: difficulties in expressive vocabulary and semantic-grammatical abilities, abstract language skills [60,62]	ADHD, anxiety, sleep disorders [63]
*NF1*	*NF1* in chromosome 17 [64]	10–40% [65,66]	n/a	5–33% mild ID [67,68]	70% mild difficulties in articulation, naming, receptive and expressive vocabulary [67]	ADHD in 38% [69,70]

## Data Availability

Not applicable.

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
