# Peer review of "Electrophysiological and Behavioral Evidence for Hyper- and Hyposensitivity in Rare Genetic Syndromes Associated with Autism"

_genes, 2022, doi:10.3390/genes13040671_

Round 1
Reviewer 1 Report
Dear Author,
Thanks for submitting your review manuscript entitled "EEG biomarkers of sensory abnormalities in rare genetic disorders associated with autism"
Before the final revision of this manuscript, the author needs to address the following comments in a scientific manner.
Major concerns:-
Please find out the following comments
• The review is incomplete.
• There are typographical errors throughout the manuscript.
• Conclusion should be added in a separate paragraph.
• References are missing. Reference style is inappropriate.
• Discussion should be improved.
• Tables should include references.
• This review is of no point without the references.
Abstract
- The abstract contains typographic errors which need to be corrected.
- Abstract is too short. It seems to be incomplete and do not provide a clear view of the data in the review.
- The author needs to correct the grammatical errors and make the abstract more understandable.
Introduction
- The reviewer feels that introduction needs many corrections and should be revised again.
- There are typographic errors like in fourth line of the 1st paragraph there is no full stop at the end, the references in the second line of the 2nd paragraph i.e. (Marco и др., 2011) (Robertson & Baron-Cohen, 2017)) should be corrected to (Marco и др., 2011; Robertson & Baron-Cohen, 2017).
- The author has not introduced or discussed about different genes involved in ASD with their specific role in causing ASD like traits and ASD like syndromes.
- The author should mention the clinical references to support the data.
Table
- The description of the table is incomplete.
- The author should add in the text about the role of the genes involved in ASD so that the genetic cause column gets justified.
- The major drawback of the table1 and table2 is that the author has not mentioned the references in it, hence it is very difficult to interpret the actuality of the data.
Description (body)
- The reference style is very hard to understand. The referencing is not done properly. It becomes difficult to interpret this review.
- Without the complete references, there is no point of this review.
Discussion
- A minimal critical analysis should be provided in discussion.
- Again, the reference style and incomplete references is a problem. The discussion needs to be checked for typographic and grammatical errors. Also, the references should be set accordingly.
Conclusion
- The conclusion of the study is missing.
- The reviewer feels that this review is totally incomplete and can’t be published.
- A major revision by is required the authors of this review.
References
- Reference list is missing.
- There is no point of this review without the references.
This reviewer considers that this paper cannot be published in the present form. This paper requires a major revision and changes in the manuscript suggested above could improve this interesting paper in a significant manner.
Reviewer 2 Report
In the current manuscript, Neklyudova et al reviewed spontaneous as well as event-related changes in brain activity in different syndromic forms of intellectual disabilities (IDs). The review is nicely written and is easy to follow. The authors have touched on a domain in ID research that many reviews choose to bypass, i.e. diverse electrophysiological underpinnings of ID and how changes in electrophysiological signatures modulate sensory processing.
I don't have any specific comments or questions on the manuscript.
Author Response
Thank you for the review!
Reviewer 3 Report
Manuscript ID: 1634981, titled as “Electrophysiological and behavioral evidence for hy-per- and hyposensitivity in rare genetic syndromes associated with autism” is an extensive information compiling different aspects including electrophysiology and behavior categorizing the Autism into hyper or hypo sensitivity. Though this study is of significant value, authors need to make few changes for this review to be complete.
1, Authors have categorized Phelan-McDermid Syndrome as hyposensitive and provided supporting human and animal studies. However, there are some discrepancies in animal studies, for example Chen et al., 2020 (reference 101) have shown the sensory hyper-reactivity. Authors might have discussed this issue in the discussion.
2, Unlike all the syndromes/diseased conditions explained in this review, authors have limited the GABA dysfunction or GABAergic role in Rett syndrome. Authors should include more studies in this aspect.
3, The final sentence in the Rett syndrome part is hard to understand (lines 493-497). I suggest authors to re-write this part.
4, Authors have clearly explained the different aspects throughout the review and during discussion. However, a brief conclusion is missing which would give reader a comprehensive understanding of the whole review and field.
Minor mistakes:
Line 113- delete the and between intellectual disability, social deficit
Line 613- repetition of “auditory” during bAEP abbreviation
In table 2- onset of epilepsy for FXS is missing
Round 2
Reviewer 1 Report
After careful revision of the manuscript, reviewer found that author successfully revised the reviewer queries.